# Seasonality of the Cardiac Biomarker Troponin in the Eastern Croatian Population

**DOI:** 10.3390/jcm7120520

**Published:** 2018-12-06

**Authors:** Mišel Mikić, Anamarija Šestak, Mile Volarić, Stjepan Rudan, Ljiljana Trtica Majnarić

**Affiliations:** 1General Hospital Vinkovci, Emergency Admission Unit Laboratory, Vladimira Gortana 25, 32100 Vinkovci, Croatia; mikic.misel@gmail.com; 2Faculty of Medicine, Josip Juraj Strossmayer University of Osijek, Josipa Huttlera 4, 31000 Osijek, Croatia; mvolaric@gmail.com; 3General Hospital Vukovar, Department of Otorhinolaryngology, Županijska ulica 35, 32000 Vukovar, Croatia; 4Department of Public Health, Faculty of Dental Medicine and Health, Crkvena 21, 31000 Osijek, Croatia; info@fdmz.hr (S.R.); ljiljana.majnaric@mefos.hr (L.T.M.); 5Department of Internal Medicine, Family Medicine and the History of Medicine, Faculty of Medicine, Josip Juraj Strossmayer University of Osijek, Josipa Huttlera 4, 31000 Osijek, Croatia

**Keywords:** high sensitive cardiac troponin assay, hospital emergency admission unit register, seasonality of cardiovascular events, cardiovascular triggering factors

## Abstract

Background: The seasonality of acute myocardial infarction and progressive heart failure has been well established so far. Cardiac troponins (cTns) are organ-specific, not disease-specific, biomarkers. The seasonality of cTns has not been reported before. Methods: Data were collected from the emergency admission unit of a community hospital in eastern Croatia for each month of the year 2014 covering the number of patients whose doctors requested high-sensitivity cTn I (hs-cTn I) testing, the number of positive test results and hospital admissions. Results: The proportion of patients with positive test results was 15.75% (350 patients out of 2221 patients referred to testing), with the males being outnumbered by the females (F: 57.15%, M: 42.85%) (*p* = 0.069). The month with the highest number of patients with positive test results was December, whereas the month with the lowest number of those patients was January (*p* < 0.001). The highest numbers of patients referred to testing (30.9%) and of those with positive test results (50.8%) were found in the oldest age group (76+). Conclusion: Tracking the results of cTns testing during patient admissions to emergency departments would be a more effective approach from a public health perspective than tracking the number of patients diagnosed with a particular cardiovascular (CV) disease and could be used as a research approach to guide a search for precipitating factors for CV disease specific to a local community.

## 1. Introduction

Seasonal variations in acute myocardial infarction (AMI) and other cardiovascular (CV) events, including hypertensive urgency and progressive heart failure (HF), have been well established so far [1,2,3]. This phenomenon has been observed in different geographic and climate regions of the world and confirmed in largescale epidemiological studies [4,5]. It has been most consistently reported among elderly people, probably due to their greater vulnerability to the influence of different environmental factors, which is due to an age-related decline in homeostatic regulation [6].

Cardiac troponin (cTn) testing has become a component of the standard diagnostic workup for patients who may have acute coronary syndrome (ACS), and it is used as a guide in making decisions for coronary interventions [7]. In the past few years, implementation of high-sensitivity cTn (hs-cTn) assays in routine diagnostics in European clinics and hospitals has significantly improved outcomes of patients with AMI [8]. The problem with this approach is that increased sensitivity of cTn assays has come at the cost of reduced specificity, thus posing new diagnostic challenges to clinicians [9]. Strategies used to improve the diagnostic accuracy of hs-cTn assays for detection of patients with AMI include the clinical probability assessment and serial hs-cTn sampling [7].

In most reports, seasonal patterns of CV events have shown peaks in the winter, even in regions with a warm climate throughout the year [10,11]. Explanations provided by the early studies were that cold weather or some other meteorological factors, such as intraday temperature oscillations, could influence these peaks [5,12]. Some studies reporting peak incidences of CV events in the summer or no seasonal variations at all have led to the general notion that extreme weather conditions, including low or high temperatures or high humidity, underlie the seasonal variations in CV events [13,14].

Knowledge accumulated over time indicates that meteorological factors cannot fully explain the seasonality of CV morbidity and mortality. Seasonal variations in other environmental factors, such as acute respiratory infections or human behaviors that can affect conventional CV factors should also be considered [15,16]. The problem with generalization of human behaviors is that large variations can occur among regions and countries due to differences in the societal, cultural and lifestyle characteristics of populations, rather than due to differences in climate and meteorological factors [17].

Insufficient knowledge of the effects of climate change and air pollution on cardiac health poses further challenges to researchers investigating the seasonality of CV events [18,19]. Notably, air pollution can vary in its effects on urban and rural areas and particular population groups [19].

Because regionally-specific societal, cultural and meteorological factors drive variations in CV events, the assessment of these factors at the community or regional levels is suggested. A promising organizational framework for this assessment is the emergency admission unit registers of a community hospital, in which measurements of cardiac biomarkers (troponins) can shed light on the seasonality of CV emergencies in the population and guide data collection on precipitating factors of CV events.

## 2. Materials and Methods

### 2.1. Study Design and Data Source

A retrospective study was conducted in the General Hospital Vinkovci. Vinkovci (35,000 inhabitants) is the administrative center of a rural area in eastern Croatia. In this region, due to the devastating economic situation, negative demographic trends and population aging have taken place in the last decades.

Data were collected from the information technology (IT) registers of the emergency admission unit for the year 2014. Cumulative data on the number of patients whose doctors requested testing on hs-cTn I and the number of patients with positive test results were estimated for each month of the year 2014 (Figure 1). For this purpose, the IT register of the emergency admission unit was screened for patients referred to the laboratory for hs-cTn I testing. Patients whose names appeared two or more times were considered repeatedly tested (Table 1). Patient test results were counted as positive if the patient had at least one positive test. Based on the 99^th^ percentile cut-off for hs-cTn I test positivity, the diagnostic limit was set as low as 40 ng/L [20].

Patients with positive hs-cTn I test results were assessed to determine whether they were admitted to the cardiology department, admitted to another hospital department, or discharged (Table 2). Patients admitted to the cardiology department were considered diagnosed with CV emergencies.

### 2.2. Patients

Patients admitted and processed at the emergency admission unit of the General Hospital Vinkovci during 2014 under suspicion of ACS were selected for analysis. They were not directly included in the study, but their outcomes (admission to the cardiology department, admission to another department or discharge) were tracked according to their referrals to the emergency laboratory for hs-cTn I testing and the results of this testing. Other information from patient records, including results of the specific diagnostic procedures, were not available without special permissions and were not used in this study.

### 2.3. Ethics Statement

The study was conducted in accordance with the Declaration of Helsinki, and the study was approved by the Ethics Committee of Faculty of Medicine, Josip Juraj Strossmayer University of Osijek (01-7929/2/17).

### 2.4. The hs-cTn Referent Test

In General Hospital Vinkovci, the hs-cTn I assay is used as the referent test for the diagnosis of ACS. This test is based on the chemiluminescence immunoassays (CLIA) analytical method and is performed on the apparatus Siemens Centaur XP (ADVIA Centaur TnI-Ultra Assay, Siemens Healthcare Diagnostics Inc., Tarrytown, NY, USA).

Analytical properties of ADVIA Centaur TnI-Ultra Assay: assay range 6–50.000 ng/L; analytical sensitivity 6 ng/L; CV 10% 30 ng/L; 99th percentile 40 ng/L.

### 2.5. Statistical Analysis

The gender-dependent, age-dependent and per month distributions of the number of requests for the hs-cTn I testing and the number of positive test results have been presented. Hi-square and Fisher`s exact tests were used to analyze differences between the distributions. A line graph, a bar graph and histograms were used to present these distributions also visually. All *p*-values were two-sided. The significance level was set to <0.05. Statistical analysis was performed using SPSS Statictical Software (version 16.0, SPSS Inc., Chicago, IL, USA).

## 3. Results

The total number of patients referred to hs-cTn I testing in 2014 in the emergency admission unit of the General Hospital Vinkovci was 2221, being almost equally distributed between men and women (47.14% and 52.86%, respectively). The proportion of patients with positive test results was 15.75% (350 patients out of 2221 tested), with the females (57.15%) outnumbering the males (42.85%) (*p* = 0.069).

The proportion of patients with positive hs-cTn I test results relative to the number of those referred to testing was constant across the months (*p* = 0.17) (Figure 1). The top month with respect to the number of patients whose doctors requested to hs-cTn I testing for both males and females was December, whereas the months with the lowest number of those patients were January and February (*p* < 0.001).

Out of the total number of 350 patients with positive hs-cTn I test results, 235 (67.14%) patients underwent repeated testing (Table 1).

Three quarters (75.14%) of patients with positive hs-cTn I test results were admitted to the cardiology department, but without a statistically significant gender-related difference (*p* = 0.77) (Table 2).

Some patients were admitted to other departments, and some were discharged from the emergency admission unit.

Most of the patients with positive hs-cTn I test results reached values of hs-cTn I that were lower than 100 (ng/L) (Table 3).

According to the number of patients whose doctors requested hs-cTn I testing and the number of patients with positive hs-cTn I test results, CV events started to occur after the age of 45 (Figure 2). There were significantly more patients referred to hs-cTn I testing and patients with positive test results in the elderly age groups (65+) (*p* < 0.001). The highest number of patients referred to testing (686 out of a total of 2221) (30.9%) and the highest number of patients with positive test results (178 out of a total of 350) (50.8%) were found in the oldest age group (76+). Only one patient who tested positive was found in the age group of 35 years or younger.

The highest number of both male and female patients with positive hs-cTn I test results were recorded in December. The lowest number of patients were recorded in January and February (Figure 3). In most months of the year, these results were higher for the females than for the males, with the exception of the summer months June and July, during which the females were outnumbered by the males, but without statistically significant gender-related differences (*p* = 0.209) (Figure 3).

According to Figure 4, the largest number of CV events can be expected in December, in the age group >65 and in particular in patients >75. Across all age groups, the frequency of CV events was lowest in January and February, followed by April and August.

## 4. Discussion

This study demonstrates, for the first time, the existence of seasonal variations in the cardiac biomarker troponin. This finding is in line with other recent studies indicating that, in addition to the classical CV risk factors, which can be significantly influenced by human behaviours, deeper pathophysiological disorders and underlying CV pathologies (e.g., insulin resistance and autonomous microcirculation regulation) show seasonal variations [21]. Oscillating patterns of these disorders could reflect complex network dysregulation due to the aging process and chronic disease development [22]. In this context, cTns can be considered not only as a biomarker of ischaemic cardiac events, but also more generally as a biomarker of the global instability of the CV system, indicating elevated risk of a wide range of CV disorders, with the most common ones being myocardiopathy, chronic renal failure and cardiac arrhythmias [20,23,24]. Given that cTns are organ-specific rather than disease-specific biomarkers, tracking results of cTn testing during the patient admission process in the emergency department would be a more meaningful approach from a public health perspective than tracking the number of diagnoses of AMI or progressive HF [25]. If performed at the community or regional level, this follow-up can provide a practical framework for more focused research on precipitating factors for CV disorders than those used in previous studies.

The high repetition rate (67.1%) for hs-cTn I testing found in this study may be due to the early hospital admissions occurring after symptoms associated with CV emergency had presented, as well as diagnostic uncertainties associated with comorbidities, the common situation in older populations such as the one examined in the present study (Table 1). Low admission rates to hospital departments other than cardiology (43 patients) (11.2%) indicate that cardiac pathology was the main reason for hs-cTn I testing (Table 2). Relatively high rates of hs-cTn I test results with values that usually indicate AMI (19.7%) may be due to the population aging and poor socioeconomic situation in eastern Croatia [20,26]. The highest number of patients referred for hs-cTn I testing and the highest number of patients with positive test results were recorded in October and December, the transition months towards colder weather, and then again in March and May to July, the transition months towards warmer weather (Figure 1). This multi-wave oscillating pattern differs from the unimodal peaks found in previous studies, in which particular CV diagnoses have been tracked. 

The rates of patients tested on hs-cTn I and those with positive test results were the highest for elderly persons (66+), which aligns with the epidemiological characteristics of CV disease (Figure 2) (Townsend). Taken together, the results of the present study suggest that the main diagnostic concern associated with repetition of hs-cTn I testing was the need to distinguish between ACS and progressive HF in elderly people. This assumption is based on evidence that HF is the main cause of hospitalization of elderly people [27]. It is especially difficult to diagnose ACS in elderly people with HF; therefore, in this context, repeated hs-cTn I testing can be useful [28]. As in this study, in previous studies in which seasonal variations of the incidence and hospitalization of progressive HF were assessed, peaks were observed in the winter, but higher rates were sometimes also observed in the summer [29].

Researchers have proposed many potential causes of the seasonal variations in ACS and HF morbidity and mortality rates. The peak prevalence of these conditions in the winter has been explained by mechanisms such as cold weather (especially short-term falls in temperature), respiratory infections and increased food intake due to the often-sedentary lifestyle during the winter [15,30,31]. The proposed mechanisms underlying the summer peaks include increased atmospheric pressure, higher relative humidity, and increased intake of sweetened beverages and fruits [29,32]. Societal schedules, including the summer work breaks for vacation, established in most European countries, and traditional celebrations, such as Christmas, may change lifestyle patterns, including traveling, eating, drinking, exercising and working habits, leading to changes in the precipitating factors of CV events [33]. These factors are often age- and gender-dependent, but they also tend to be specific to and constant within regional communities. For example, the main characteristics of communities in the eastern Croatian region are population aging, poor socioeconomic conditions and rural household activities, along with a continental climate (indicating extreme seasonal variations in weather conditions). The healthcare system in this region, although well organized, is characterized by a low level of CV preventive and public health measures.

Researchers investigating precipitating factors for CV events increasingly emphasize the effects of multiple, concurrently acting factors [16]. Some of these scenarios are likely to be suggested by the results of the present study. Age- and gender-related seasonal trends in cardiac troponins, estimated from the data available in the community hospital emergency unit registers, can guide evaluation of these factors.

According to our results, positive results for cTn testing, which indicate CV events, are rare in patients younger than middle aged (45 or younger). After 45, the frequency of CV events steadily increases with age (Figure 2). The month with the largest increase in positive hs-cTn I test results was December, whereas the lowest numbers of positive results were recorded in January and February, followed by April and August (Figure 1 and Figure 3). The peak in December can be explained by the effect of Christmas and other traditional celebrations taking place during this month. An alternative explanation is more changeable weather, with short-term fluctuations in temperature, which in the continental climate regions is more typical for December than for January or February. These seasonal changes have been especially intense in recent years, probably due to global warming. The latter explanation seems more plausible than the former because in contrast to December, April (the month of Easter celebrations) showed a record-low in positive results.

Another potential cause of the peaks observed in late autumn’s colder months, October and December, in particular when the oldest patient group (76+) is considered, might be the susceptibility of elderly people to respiratory infections (Figure 4). On the contrary, low rates of hs-cTn I positive results may have been recorded in the winter months (January and February) because the rate of influenza vaccination for elderly people in the studied region is usually fairly high (Figure 3 and Figure 4) [34]. Our results also suggest another hidden factor as a possible cause of marked variations in the levels of cTns during the winter months, one not mentioned in previous studies or reflected in weather reports: the difference in darkness and light between months at the end and the beginning of the year [35].

The inclusion of gender-related factors in an analysis of seasonal variations in positive cTn test results has revealed further precipitating factors of CV events. As suggested by Figure 3, male patients more often test positive for cTn in the beginning of the summer (in June and July), which probably reflects men’s greater sensitivity to environmental challenges, such as overheating during hot weather. Alternatively, there could be gender-related differences in exposure to outdoor activities that are usually more common in the beginning of the summer. Household activities are proposed as potential CV precipitation factors based on the characteristics of the eastern Croatian local population, which include elderly (not occupationally active) people and traditional ways of life in a rural area. Male patients’ higher rate of CV events during the early summer peak can also be explained by the gender-related bias for development of specific CV entities, such as the association between the male gender and the ST-elevation type of AMI (Figure 3) [36]. Yet, these considerations require further evaluation due to the small number of patients evaluated each month in this study.

As suggested by Figure 4, all age groups show similar seasonal patterns in positive cTn testing results, with only slight fluctuations. The results indicate common seasonal patterns in emergent CV conditions, which could be specific to the population of the eastern Croatian region. The uniformity of patterns across age ranges could reflect population aging and the distribution bias of the positive cTn test results towards the older age groups. These patterns include the multiple oscillating peaks, taking place in the spring, at the beginning of the summer and during autumn’s cold months, with nadirs in January, February, April and August. However, these results must be interpreted cautiously because of the small number of patients evaluated per month and the limited follow-up time of one year.

## 5. Conclusions

Following cardiac biomarker troponin testing results in the emergency admission unit of a community hospital can provide quick insight into the age- and gender-dependent seasonal variations of global CV emergencies, the patterns of which are typical for the local population. These identified patterns, together with knowledge of the societal and cultural characteristics of the local population, and information on meteorological and weather conditions, can direct research on CV precipitation factors that are considered relatively constant for a certain region. Tracking the levels of the cardiac biomarker troponin provides a global view of CV emergencies within a given population, and this approach is more rational from a public health perspective than tracking the incidence of a particular form of CV disease.

## Figures and Tables

**Figure 1 jcm-07-00520-f001:**
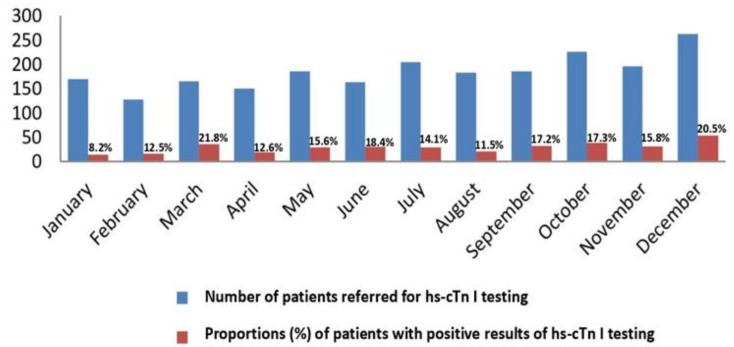
Proportions (%) of positive results of the hs-cTn I test relative to the number of requests. A distribution by months.

**Figure 2 jcm-07-00520-f002:**
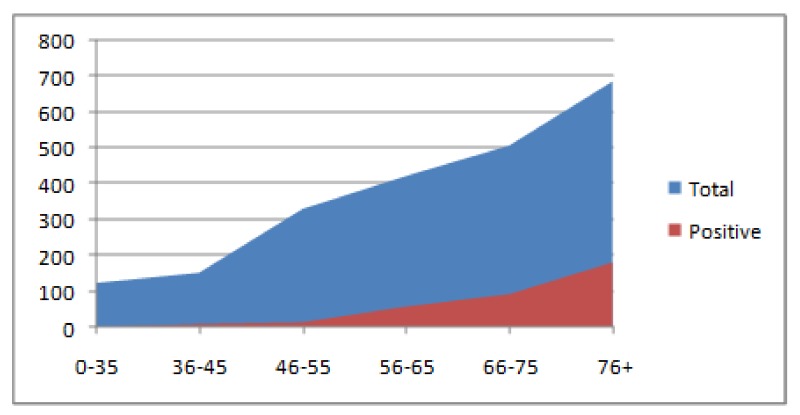
The proportion of positive results of the hs-cTn I test relative to the total number of requests. A distribution according to the six age groups.

**Figure 3 jcm-07-00520-f003:**
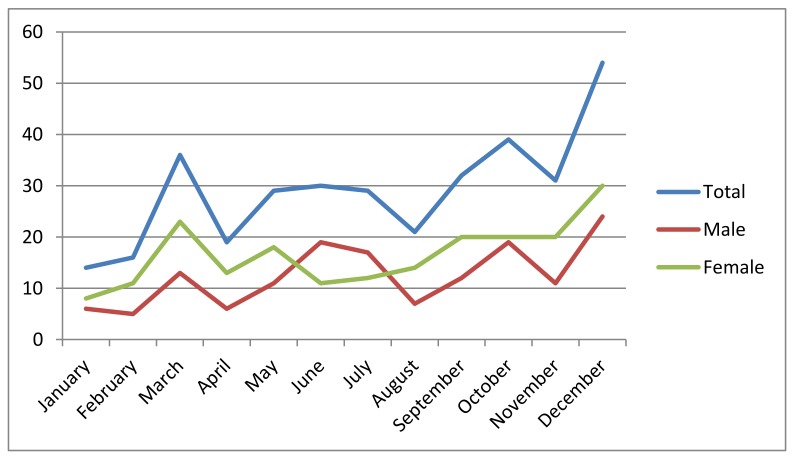
Positive results of the hs-cTn I test, separately presented for males and females. A distribution by months.

**Figure 4 jcm-07-00520-f004:**
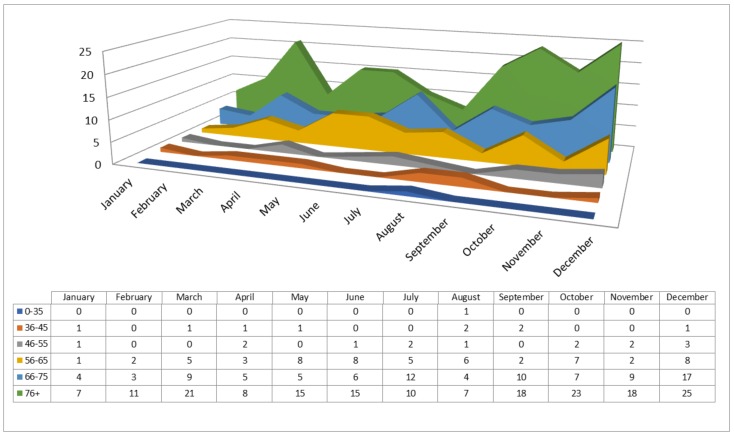
Positive results of the hs-cTn I test distributed by months and the six age groups.

**Table 1 jcm-07-00520-t001:** The hs-cTn I test repetition rates.

The Number of Patients with Repeated hs-cTn I Tests	*n* (%)	*p*-Value
1	92 (26.3)	
≥2	143 (40.9)	0.06
0	115 (32.9)	
The number of patients who tested positive	350 (100.0)	

**Table 2 jcm-07-00520-t002:** Hospital admission rates of patients who tested positive on hs-cTn I and gender-related distribution.

Department	Males/*n* (%)	Females/*n* (%)	Total
Cardiology	106 (73.1)	157 (76.6)	263
Pulmonology	15 (10.3)	14 (6.8)	29
Internal medicine	4 (2.8)	5 (2.4)	9
Infectology	1 (0.7)	2 (1.0)	3
Neurology	0 (0.0)	2 (1.0)	2
Discharged	19 (13.1)	25 (12.2)	44
Total	145 (100.0)	205 (100.0)	350

**Table 3 jcm-07-00520-t003:** Values of hs-cTn I in patients with positive hs-cTn I test results.

Range of hs-cTn I Values (ng/L)	Patients *n* (%)
40 < 100	281 (80.3)
100–5000 and >	69 (19.7)
Total	350 (100.0)

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
