# Peer review of "Seasonality of the Cardiac Biomarker Troponin in the Eastern Croatian Population"

_jcm, 2018, doi:10.3390/jcm7120520_

Reviewer 1 Report

In the current manuscript, the authors explain the relationship between seasonal variation, cardiac biomarkers in a selected population. Despite the great efforts, the findings are not novel. The findings has been described before. And based on biomarkers to distinguish between ACS and heart failure solely will be fair for clinical application or research. I feel the paper need to re address the primary outcomes. 

Author Response

Response to Reviewer 1 Comments

Point 1:  Moderate English changes required

Response 1: Thank you for a very useful comment and suggestion. We have fully accepted the remark and manuscript text has been checked by professional English editing service.

Point 2: Does the introduction provide sufficient background and include all relevant references? –Not applicable.

Response 2: Thank you for a very useful comment and suggestion. We have fully accepted the remark and changed the text of introduction (highlited in manuscript) and added some new references (also highlited in manuscript).

Point 3: Is the research design appropriate?-Must be improved

Response 3: Thank you for a very useful comment and suggestion. We have fully accepted the remark and added some more information about study design (highlited in manuscript).

Point 4: Are the methods adequately described?-Must be improved

Response 4: Thank you for a very useful comment and suggestion. We have fully accepted the remark and added some more information about study design, data source and patients (highlited in manuscript).

Point 5: Are the results clearly presented?-Must be improved

Response 5: Thank you for a very useful comment and suggestion. We have fully accepted the remark and changed the order of the text in Results section to be more clear and we have changed the appearance of the Table1. and Table 2. To be more clear. All changes are highlited in manuscript.

Point 6: Are the conclusions supported by the results?- Can be improved.

Response 6: Thank you for a very useful comment and suggestion. We have fully accepted the remark and changed the text of conclusion and text now reads:˝ Following cardiac biomarker troponin testing results in the emergency admission unit of a community hospital can provide quick insight into the age- and gender-dependent seasonal variations of global CV emergencies, the patterns of which are typical for the local population. These identified patterns, together with knowledge of the societal and cultural characteristics of the local population and information on meteorological and weather conditions, can direct research on CV precipitation factors that are considered relatively constant for a certain region. Tracking the levels of the cardiac biomarker troponin provides a global view of CV emergencies within a given population, and this approach is more rational from a public health perspective than tracking the incidence of a particular form of CV disease.˝

All changes in text are highlited in manuscript.

Point 7: In the current manuscript, the authors explain the relationship between seasonal variation, cardiac biomarkers in a selected population. Despite the great efforts, the findings are not novel. The findings has been described before. And based on biomarkers to distinguish between ACS and heart failure solely will be fair for clinical application or research. I feel the paper need to re address the primary outcomes. 

Response 6: Thank you for a very useful comment and suggestion. We have fully accepted the remark and changed some of the text of discussion and highlited all changes.

Reviewer 2 Report

The authors present an article on „The high sensitive cardiac troponin assay register as a basis for research on the seasonality of the  cardiovascular triggering factors in older people”. While the topic is of interest, the manuscript’s structure should be reworked.

Please revise the title and re-type.

In the abstract, please include eligibility criteria, how many patients were admitted in the Emergency Admission Unit in 2014?

In the section Materials and Methods, please include details on patient characteristics

Please revise and correct accordingly, the section Discussion, acute IM what does it mean?

Author Response

Response to Reviewer 2 Comments

Point 1:  Moderate English changes required

Response 1: Thank you for a very useful comment and suggestion. We have fully accepted the remark and manuscript text has been checked by professional English editing service.

Point 2: Does the introduction provide sufficient background and include all relevant references? –Must be improved.

Response 2: Thank you for a very useful comment and suggestion. We have fully accepted the remark and changed the text of introduction (highlited in manuscript) and added some new references (also highlited in manuscript).

Point 3: Is the research design appropriate?-Must be improved

Response 3: Thank you for a very useful comment and suggestion. We have fully accepted the remark and added some more information about study design (highlited in manuscript).

Point 4: Are the methods adequately described?- Can be improved

Response 4: Thank you for a very useful comment and suggestion. We have fully accepted the remark and added some more information about study design, data source and patients (highlited in manuscript).

Point 5: Are the results clearly presented?-Must be improved

Response 5: Thank you for a very useful comment and suggestion. We have fully accepted the remark and changed the order of the text in Results section to be more clear and we have changed the appearance of the Table1. and Table 2. to be more clear. All changes are highlited in manuscript.

Point 6: Are the conclusions supported by the results?- Can be improved.

Response 6: Thank you for a very useful comment and suggestion. We have fully accepted the remark and changed the text of conclusion and text now reads:˝ Following cardiac biomarker troponin testing results in the emergency admission unit of a community hospital can provide quick insight into the age- and gender-dependent seasonal variations of global CV emergencies, the patterns of which are typical for the local population. These identified patterns, together with knowledge of the societal and cultural characteristics of the local population and information on meteorological and weather conditions, can direct research on CV precipitation factors that are considered relatively constant for a certain region. Tracking the levels of the cardiac biomarker troponin provides a global view of CV emergencies within a given population, and this approach is more rational from a public health perspective than tracking the incidence of a particular form of CV disease.˝

All changes in text are highlited in manuscript.

Point 7: The authors present an article on „The high sensitive cardiac troponin assay register as a basis for research on the seasonality of the  cardiovascular triggering factors in older people”. While the topic is of interest, the manuscript’s structure should be reworked.

Response 6: Thank you for a very useful comment and suggestion. We have fully accepted the remark and changed the title of manuscript into ˝Seasonality of the cardiac biomarker troponin in the eastern Croatian population˝, we have changed some of the text of manuscript in all sections (highlited in manuscript).

Point 8: Please revise the title and re-type.

Response 6: Thank you for a very useful comment and suggestion. We have fully accepted the remark and changed the title of manuscript into ˝Seasonality of the cardiac biomarker troponin in the eastern Croatian population˝.

Point 9: In the abstract, please include eligibility criteria, how many patients were admitted in the Emergency Admission Unit in 2014?

Response 6: Thank you for a very useful comment and suggestion. We have fully accepted the remark and added information in abstract (highlited in manuscript):

˝ Data were collected from the emergency admission unit of a community hospital in eastern Croatia for each month of the year 2014 covering the number of patients whose doctors requested high-sensitivity cTn I (hs-cTn I) testing, the number of positive test results and hospital admissions.˝

There were 2,221 patients in total that were admitted in the Emergency Admission Unit in 2014.

Point 10: In the section Materials and Methods, please include details on patient characteristics

Response 6: Thank you for a very useful comment and suggestion. We have fully accepted the remark and added some text in section Material and Methods (highlited in manuscript). The text:˝ Patients admitted and processed at the emergency admission unit of the General Hospital Vinkovci during 2014 under suspicion of ACS were selected for analysis. They were not directly included in the study, but their outcomes (admission to the cardiology department, admission to another department or discharge) were tracked according to their referrals to the emergency laboratory for hs-cTn I testing and the results of this testing. Other information from patient records, including results of the specific diagnostic procedures, were not available without special permissions and were not used in this study.˝

Point 11: Please revise and correct accordingly, the section Discussion, acute IM what does it mean?

Response 6: Thank you for a very useful comment and suggestion. We have fully accepted the remark and revised the discussion section (highlited in manuscript).

Acute IM has been changed in acute myocardial infarction AMI.

Reviewer 3 Report

Quite honestly I do not understand why following troponin results would be more meaningful than to simply follow the numbers of coronary syndrome and heart failure respectively.

There are more factors that might contribute to seasonal variation in cardiovascular morbidity. In most European countries there is a seasonal work cycle with vacations and more relaxed conditions during the summer part of the year. Of course that does not explain the low numbers of troponin bursts reported for January and February. But as the authors point out there are several factors of potential importance so one factor does not explain all the variation.There is an astronomical cycle with variations in darkness and light – not reflected in weather reports. 

But the authors do present an interesting discussion regarding an array of factors that could contribute to seasonal variations in cardiovascular morbidity, such as Christmas, violent variations in weather (climatic change) etc. A problem, however, is that they discuss many parameters and this in combination with pattern differences related to age and gender make it statistically difficult to draw conclusions. The numbers of subjects in the different boxes are quite small in some places. The authors have to make a cautionary statement about the risk of mass significance in their discussion

There are number of language problems, for instance

Vinkovci, Eastern Croatia. In this region, the prevalence of CV disease overcomes….

I  presume the authors mean …exceeds… rather than overcomes

In several places they say …have showed…  The text needs professional language editing

At least on my screen the legend for figure 1 is incomplete and confusing. For instance, do the percentages in the figure refer to red or blue? I presume 20.5 is the percentage red/blue but that is not obvious from the legend.

Author Response

Response to Reviewer 3 Comments

Point 1: Extensive editing of English language and style required

Response 1: Thank you for a very useful comment and suggestion. We have fully accepted the remark and manuscript text has been checked by professional English editing service.

Point 2: Does the introduction provide sufficient background and include all relevant references? –Can be improved.

Response 2: Thank you for a very useful comment and suggestion. We have fully accepted the remark and changed the text of introduction (highlited in manuscript) and added some new references (also highlited in manuscript).

Point 3: Is the research design appropriate?-Can be improved

Response 3: Thank you for a very useful comment and suggestion. We have fully accepted the remark and added some more information about study design (highlited in manuscript).

Point 4: Are the results clearly presented?-Can be improved

Response 5: Thank you for a very useful comment and suggestion. We have fully accepted the remark and changed the order of the text in Results section to be more clear and we have changed the appearance of the Table1. and Table 2. to be more clear. All changes are highlited in manuscript.

Point 5: Are the conclusions supported by the results?- Can be improved.

Response 6: Thank you for a very useful comment and suggestion. We have fully accepted the remark and changed the text of conclusion and text now reads:˝ Following cardiac biomarker troponin testing results in the emergency admission unit of a community hospital can provide quick insight into the age- and gender-dependent seasonal variations of global CV emergencies, the patterns of which are typical for the local population. These identified patterns, together with knowledge of the societal and cultural characteristics of the local population and information on meteorological and weather conditions, can direct research on CV precipitation factors that are considered relatively constant for a certain region. Tracking the levels of the cardiac biomarker troponin provides a global view of CV emergencies within a given population, and this approach is more rational from a public health perspective than tracking the incidence of a particular form of CV disease.˝

All changes in text are highlited in manuscript.

Point 6: The numbers of subjects in the different boxes are quite small in some places.

Response 6: Thank you for a very useful comment and suggestion. We have fully accepted the remark and we have revised all Tables and Figured to be more clear and visible.

Point 7: The authors have to make a cautionary statement about the risk of mass significance in their discussion

Response 6: Thank you for a very useful comment and suggestion. We have fully accepted the remark and we have revised the text of discussion section. (highlited in text)

Point 9: There are number of language problems

Response 6: Thank you for a very useful comment and suggestion. We have fully accepted the remark and manuscript text has been checked by professional English editing service.

Point 10: Figure 1 is incomplete and confusing. For instance, do the percentages in the figure refer to red or blue? I presume 20.5 is the percentage red/blue but that is not obvious from the legend.

Response 6: Thank you for a very useful comment and suggestion. We have fully accepted the remark and we have changed the Figure 1. to be more clear.

Round  2

Reviewer 1 Report

In the current revision, the authors address all the previous comments and no further revisions needed

Reviewer 3 Report

I have done the review and I now recommend publication without any changes.